# Protein and Leucine Intake at Main Meals in Elderly People with Type 2 Diabetes

**DOI:** 10.3390/nu15061345

**Published:** 2023-03-10

**Authors:** Elena Massimino, Anna Izzo, Carmen Castaldo, Anna Paola Amoroso, Angela Albarosa Rivellese, Brunella Capaldo, Giuseppe Della Pepa

**Affiliations:** 1Department of Clinical Medicine and Surgery, Federico II University, Via Sergio Pansini 5, 80131 Naples, Italy; 2Institute of Clinical Physiology, National Research Council-CNR, Via Giuseppe Moruzzi 1, 56124 Pisa, Italy

**Keywords:** protein intake, leucine intake, meal, elderly, type 2 diabetes

## Abstract

Background: The recommended protein intake for the elderly is 25–30 g at main meals, with at least 2500–2800 mg of leucine at each meal. There is still little evidence regarding the amount and distribution of protein and leucine intake with meals in the elderly with type 2 diabetes (T2D). In this cross-sectional study, we evaluated protein and leucine intake at each meal in elderly patients with T2D. Methods: A total of 138 patients (91 men and 47 women) with T2D, aged 65 years or older, were included. Participants performed three 24-h dietary recalls for the evaluation of their dietary habits and protein and leucine intake at meals. Results: The average protein intake was 0.9 ± 0.2 g/kg body weight/day, and only 23% of patients complied with the recommendations. The average protein intake was 6.9 g at breakfast, 29 g at lunch, and 21 g at dinner. None of the patients reached the recommended protein intake at breakfast; 59% of patients complied with the recommendations at lunch; and 32% at dinner. The average leucine intake was 579 mg at breakfast, 2195 g at lunch, and 1583 mg at dinner. The recommended leucine intake was not reached by any patient at breakfast, by 29% of patients at lunch, and by 13% at dinner. Conclusions: Our data show that, in elderly patients with T2D, the average protein intake is low, particularly at breakfast and dinner, and that leucine intake is remarkably lower than the recommended levels. These data raise the need to implement nutritional strategies capable of increasing protein and leucine intake in the elderly with T2D.

## 1. Introduction

The advancements in social, medical, and technological fields that occurred in the last decade had considerably increased the longevity of the population, highlighting the process of aging as the main responsible for the progressive impairment in all body districts [1].

Age-related changes involve mainly skeletal muscle and are characterized by a progressive loss of mass, strength, and performance that greatly affect the physical function of elderly people, promoting frailty and disability [2]. Population aging has also been observed in people with type 2 diabetes (T2D) [3], in whom a higher risk of frailty, sarcopenia, and disability has been reported [4,5,6].

Large evidence shows that protein intake is directly related to muscle mass, strength, and function in elderly people [7,8] and could have a prominent role in preventing sarcopenia and frailty [9,10,11,12].

It is important to consider that elderly people have the so-called “anabolic resistance”, which is defined as the reduced ability of skeletal muscle to increase protein synthesis in response to anabolic stimuli, such as dietary protein and physical activity [13,14,15]. This condition accelerates the age-related loss of muscle mass and function, thus contributing to functional limitations and disability in elderly people. Although there is no univocal view on which is the best strategy to mitigate anabolic resistance in the elderly, there is general agreement that increasing protein intake may be effective [13,14,15]. Thus, an average daily intake of 1–2 g/kilogram body weight (BW) is recommended and even higher in elderly people with or at risk of malnutrition [16,17]. Despite these indications, in real life, the dietary protein intake in the elderly population is very far from that recommended [18]. Previous studies have shown that also the distribution of protein intake at mealtime could positively impact muscle health in older people, with evidence indicating an intake of 25–30 g of protein at each meal as adequate to promote skeletal muscle synthesis [16,19,20]. Lastly, the quality of protein, the source (animal or plant), and the amino acid content, particularly the branched-chain amino acids, primarily leucine, are equally important for the skeletal muscle’s health [21,22]. As known, leucine beneficially acts on protein anabolism through the enzymatic activation of the Mammal Target of Rapamycin (mTOR), a serine-threonine protein kinase involved in the regulation of cellular growth and protein turnover by promoting protein synthesis and inhibiting proteolysis [21,22]. Evidence-based recommendations indicate a leucine intake of 3000 mg at each meal in elderly people [16].

Protein turnover at the level of skeletal muscle is regulated by several factors, including the digestion of food proteins and the subsequent absorption of amino acids, the release of post-prandial insulin, the threshold of muscle anabolism, and the microvascular perfusion of the tissue [13]. Most of these processes can be greatly affected by the presence of diabetes, especially if there is poor metabolic control. Elderly people with diabetes, in fact, present a greater risk of loss of muscle mass and strength and, consequently, a higher rate of physical disability [23]. Based on these considerations, they can greatly benefit from an adequate protein intake.

There is little evidence on the adherence of elderly people with T2D to the recommended protein intake; the available data indicate that most of them do not meet the recommended protein intake, have a lower nutrient density, and are more likely to skip meals [24,25]. Furthermore, at present, there is no information on the daily protein distribution and leucine intake at the main meals in T2D.

In view of the important role of protein intake in relation to muscle mass, strength, and function in elderly people and considering the importance of the content of branched-chain amino acids, leucine in particular for its powerful effect on muscle growth [16,21,22], the aim of the present study is to provide additional information on the daily protein and leucine intake in elderly patients with T2D. In-depth knowledge on this topic could help maintain muscle health and function, prevent sarcopenia and frailty, and promote healthy dietary strategies in T2D.

## 2. Materials and Methods

### 2.1. Study Design and Participants

We conducted a cross-sectional study in elderly patients with T2D who underwent the yearly complication assessment visit at the Diabetes Unit of Federico II University of Naples from June 2020 to September 2020. The inclusion criteria were T2D, age ≥ 65 years, and the ability to answer questionnaires during phone interviews. The exclusion criteria were eating disorders, chronic diabetic complications that could interfere with food intake (i.e., gastroparesis and atrophic gastritis), history of alcoholism and substance abuse, supplementation with vitamins, nutraceuticals, or antioxidants, and any other acute/chronic disease severely affecting health status and limiting data collection. Patients who met the inclusion criteria underwent, between April 2021 and July 2021, questionnaires during phone interviews by expert dieticians for the assessment of dietary habits.

The study protocol, performed in accordance with the Declaration of Helsinki, was approved by the Ethics Committee of Federico II University, and all participants provided written informed consent to the use of their clinical and laboratory data and for being included in the study.

### 2.2. Assessment of Clinical and Biochemical Parameters

During the yearly complication assessment visit, participants underwent clinical evaluation and the collection of a fasting blood sample. Body weight was measured by mechanic balance (Seca 709), height with a bar-altimeter, and waist circumference by anelastic meter. All measurements were taken with accuracy to the nearest 0.1 kg and 0.1 cm, with the patient wearing light clothing and no shoes. Body mass index (BMI) was calculated as body weight in kg divided by the square of body height in meters. All participants underwent a complete screening for chronic complications according to a standardized protocol, including a clinical examination and a dilated eye exam for diabetic retinopathy screening. Nephropathy was evaluated by the assessment of urinary albumin excretion rate, serum creatinine, and eGFR. Autonomic nerve function was examined by standardized cardiovascular reflex tests: parasympathetic function was evaluated by the heart rate variability through a deep breathing test (beat-to-beat variation test), and sympathetic function was assessed by the blood pressure response to standing. Peripheral neuropathy was evaluated with a bilateral vibration perception examination, a tactile perception test with the Semmes-Weinstein monofilament, and ankle reflex assessments.

Data on comorbidities and drugs taken were collected from medical records.

Physical activity was investigated by the International Physical Activity Questionnaire short form (IPAQ-SF) [26]. The IPAQ-SF consisted of questions on the physical activity carried out in the last 7 days, considering: (1) vigorous-intensity activity such as aerobics; (2) moderate-intensity activity such as leisure cycling; (3) walking; and (4) sitting [26]. The level of physical activity was classified into three categories: low, moderate, and high. From these values, we calculated the total number of physical activities per week and the total amount of physical exercise in terms of metabolic equivalent tasks (MET) by multiplying durations, frequencies, and MET scores for each type of activity. Furthermore, three possible categories were classified: inactive (<700 MET × week), moderately active (700–2500 MET × week), and active (>2500 MET × week), according to the scoring system provided by the IPAQ scoring protocol [26].

Plasma glucose, total cholesterol, and triglycerides were analyzed by the colorimetric method. High-density lipoprotein (HDL) cholesterol was analyzed by the precipitation method, and low-density lipoprotein (LDL) cholesterol was calculated by Friedwald’s formula. Glycated hemoglobin (HbA1c) was measured by the HPLC method. All biochemical analyses were performed in a central laboratory.

### 2.3. Assessment of Dietary Habits

Three 24-h dietary recalls on two weekdays and one weekend day, administered by trained dietitians and collected by validated phone interviews [27,28], were used to assess dietary habits and estimate daily protein and leucine intake at meals, considering the following eating times: breakfast, lunch, dinner, and in between the main meals. During the 24 h dietary recall, the dietitians registered all the meals and beverages consumed at any time during the previous day. Patients were asked to detail their food intake, weigh their food items, or describe the food amounts in standard household measurements. The energy, macronutrient, and branched-chain amino acid intakes were calculated by means of the average of three 24-h dietary recalls using the nutritional software MetaDieta (Meteda s.r.l., Ascoli-Piceno).

The total daily protein intake was expressed as g/day and g/kg BW/day, leucine intake as mg/day, and was compared with the dietary recommendations for the elderly population [16]. An adequate intake of protein and leucine for meals was considered to be greater than 25 g and 2500 mg for protein and leucine, respectively. The energy intake values outside the range of 1000–6000 kcal/day were considered unviable.

### 2.4. Statistical Analysis

Data are presented as the mean ± standard deviation for continuous variables or as frequencies and percentages for categorical variables. Data on energy, macronutrients, and leucine intake are presented by gender, and differences in a continuous variable between men and women were tested using the independent samples t-test. A *p* value < 0.05 was considered statistically significant. Statistical analyses were performed using SPSS 26.0 software (SPSS/PC; IBM, Armonk, NY, USA).

## 3. Results

### 3.1. Characteristics of the Participants

The main clinical and anthropometric characteristics of the participants (91 men and 47 women, respectively) are reported in Table 1. The mean age was 72 ± 4 years, the mean BMI was 29.4 ± 4.7 kg/m^2^, and 42% of patients were obese. Moreover, the majority of our patients were inactive. Participants had good glucose control and a high prevalence of hypertension and dyslipidemia, which were well controlled with pharmacologic therapy. Cardiovascular disease was present in one-third of our cohort, while signs of microvascular complications were present in ~23% of the patients (Table 1).

### 3.2. Daily Energy and Nutrient Intake of the Study Population

The average daily energy and nutrient intake are reported in Table 2. As expected, women had a significantly lower daily energy intake compared with men (1322 ± 320 vs. 1462 ± 345 Kcal/day, respectively; *p* = 0.022). A high percentage of participants (77%) did not adhere to the recommended dietary protein intake expressed as g/kg BW/day, with women having a higher protein intake than men (1.0 ± 0.3 vs. 0.9 ± 0.2 g/kg BW/day, respectively; *p* = 0.029). With regard to the other macronutrients and branched-chain amino acids, no differences were observed between sexes (Table 2).

### 3.3. Daily Energy, Macronutrient, and Leucine Intake at the Main Meals Consumed by the Study Population

The energy and macronutrient intake were higher at lunch (Table 3); women had a significantly lower intake of fat and carbohydrates at lunch compared with men (Table 3).

With regard to protein intake, the average intake was 6.9 ± 3.7 g at breakfast, 29 ± 10 g at lunch, and 21 ± 10 g at dinner. None of the participants reached the recommended protein intake at breakfast; 59% of them complied with the recommendations at lunch, and 32% at dinner (Figure 1).

With regard to leucine intake, the average intake was 579 ± 334 mg at breakfast, 2195 ± 1199 mg at lunch, and 1583 ± 818 mg at dinner. None of the patients reached the recommended leucine intake at breakfast; 29% met the recommendation at lunch; and 13% at dinner (Figure 1). Approximately half of the patients reported having one snack between the main meals.

The main sources of protein intake were represented by cereals (25%), meat and poultry (24%), dairy products (19%), legumes (19%), and fish (12%) while the main sources of leucine intake were represented by cereals (29%), meat and poultry (25%), dairy products (21%), fish (13%), and legumes (9%).

## 4. Discussion

In this cross-sectional study, we investigated the dietary habits of elderly patients with T2D, focusing on the daily protein and leucine intake at the main meals. Our data indicate that: (1) the mean daily protein intake is lower than that recommended, and its distribution with meals does not meet recommendations (none of the patients comply with the recommended level at breakfast, 59% at lunch, and 32% at dinner); (2) the leucine intake is markedly below the recommendations, and none of the patients achieve the recommended intake at breakfast, 29% at lunch, and 13% at dinner.

There is evidence that a lower protein intake is associated with sarcopenia and frailty [9,10,11,12], which might increase the risk of mortality in patients with T2D [29,30].

Despite this evidence, daily dietary protein intake in the elderly population is very far from the recommendation (1–2 g/kg BW) [16,17], and accordingly, in our population, we found that only 23% of patients adhered to the protein recommendation.

In addition to assessing total protein intake, great attention has been paid to the distribution of dietary protein at the main meals. The available evidence indicates that approximately 25–30 g of protein per meal are required to positively affect muscle health in older people [16,19,20]. In our patients, the protein intake is remarkably below the recommended goal at breakfast and dinner, and only 59% of the patients achieve it at lunch. To the best of our knowledge, this is the first study assessing protein distribution at meals in elderly patients with T2D. Interestingly, the present findings are in line with a previous study by Roousset et al. who reported a similar meal distribution of protein intake in elderly people without diabetes (14% at breakfast, 56% at lunch, and 28% at dinner) [31].

An unbalanced distribution of dietary protein with meals was also reported in community-dwelling, frail, and institutionalized Dutch elderly people, in whom the highest amount of protein was consumed at dinner, i.e., 24–31 g, while it was 10–12 g at breakfast and 15–23 g at lunch [32]. These data were confirmed in elderly U.K. people in whom the proposed dietary protein threshold (20–30 g per meal) was achieved by 7.5%, 7.5%, and 30% of patients at breakfast, lunch, and dinner, respectively [33]. Similarly, in elderly people from the U.S.A., the amount of dietary protein was 13 g at breakfast, 17 g at lunch, and 30 g at dinner [34]. Similar findings were obtained in a Japanese population [35]. Based on the available evidence, breakfast appears to be the meal with the lowest protein intake (data confirmed also by the European Seneca study showing that in Italy the average percentage of calorie intake at breakfast is limited and lower compared to other European countries [36], and rarely high in proteins [37]), while the maximum protein intake generally occurs at dinner; on the contrary, in our Mediterranean population the highest intake was observed at lunch, probably due to traditional and cultural differences among countries; however, this difference does not seem to affect the association between meal protein intake and muscle health [38], considering that also lifestyle factors and physical activity are predictors of muscle health independently of the daily protein distribution [39].

The current guidelines refer to a mixed protein intake, derived from both animal and vegetable sources. It is well recognized that, in addition to the quantitative aspect, it is important to take into account the quality of dietary proteins, which mainly depends on their bioavailability (linked to digestion and absorption processes) and biological value (amino acid profile). Both parameters are higher in proteins of animal origin than in those of vegetable origin. In fact, the bioavailability (digestibility and absorption) of animal proteins ranges from 90% to 99%, while that of vegetable proteins ranges from 70% to 90%. [40]. The high biological value of animal proteins is determined by their content and optimal proportions of essential amino acids, while the proteins present in plant foods have a suboptimal amino acid profile because they are deficient in one or more essential amino acids, a deficit that can be filled by combining plant foods of different origins.

In our patients, the average intake of vegetal protein was approximately 43% (25 ± 7.2 g/day) of the daily protein intake (57 ± 16 g/day), and there were no significant differences between men and women.

The high biological value of proteins is known to optimize skeletal muscle metabolism, especially in elderly people [41]. More in-depth studies are needed to evaluate the differential impact of proteins from various animal and vegetable sources on muscle health and function [42].

With regard to dietary amino acid composition, great attention has been paid to branched-chain amino acids, primarily leucine, for their powerful effect on muscle growth.

In view of the important role that the content of branched-chain amino acids, in particular leucine, plays in promoting the synthesis of muscle proteins [16,21,22,43], in the present study we specifically evaluated leucine intake at each meal. In our patients, the average daily intake of leucine was ~4300 mg, a value significantly lower than that recommended; interestingly, none of the patients met the threshold at breakfast, 29% reached it at lunch, and 13% reached it at dinner. These data are in accordance with studies performed in German elderly people, in whom 14% of men and 4% of women reached two meals providing 2500 mg of leucine every day [44]. Similarly, in an Asian population, Ishikawa-Takata et al. showed that almost all participants did not meet the recommended level of leucine intake at breakfast, and half to three-quarters met the recommended levels at lunch [35].

The major contributors to the daily leucine intake are foods of both vegetable and animal origin, the main sources being cereals, meat, fish, and cheese.

Our findings expand the literature on this important topic, demonstrating the inadequacy of protein and leucine intake in the elderly with T2D. In light of the higher risk of frailty, disability, and sarcopenia in elderly patients with T2D compared with their nondiabetic counterparts [4], the present data underline the need to implement appropriate nutritional preventive strategies.

Notably, the daily energy intake in our population results in being lower than that recommended [45], particularly in women. This finding is in line with other studies performed in elderly with T2D [46,47,48] and could be related to the energy restriction prescribed in T2D patients according to their dietary treatment, although a possible underreporting of food intake due to the limitation of the 24-h dietary recall cannot be excluded. In any case, to increase energy content and protein intake, patients should be encouraged to consume cereals, legumes, meat and poultry, dairy products, and fish, particularly at breakfast and dinner [49]. This nutritional approach could help achieve an adequate daily protein intake, maintain a balanced protein distribution during the day [50,51], and, equally important, have a beneficial impact on postprandial glucose control [52,53].

It is important to underline that muscle health is also influenced by physical activity, and, in our population, the proportion of inactive patients was quite high (64%). In this regard, evidence shows that protein supplementation increases muscle mass and handgrip strength in older adults only when combined with resistance exercise [54], suggesting that in elderly people, according to functional status, other determinants such as regular resistance exercise and reduced sedentary time might have a positive impact on muscle health beyond adequate protein intake [55].

Our study has some limitations that should be acknowledged. First, the cross-sectional nature of the study does not allow for the establishment of a cause-and-effect relationship, and the relatively small sample size make it difficult to draw strong conclusions. Second, the limitation of the 24-h dietary recall related to day-to-day variations arising from within-subject random error should also be considered. To overcome this limitation, we administered 24-h dietary recalls three times. Furthermore, the limitations of the phone interview should be considered. However, previous studies showed that a well-structured and expertly managed 24-h dietary recall by phone can provide an accurate estimate of the macronutrient intake [28]. Third, we do not have data on serum leucine levels. Fourth, our data refer to a single study site in southern Italy, which may not be representative of the whole population of elderly with T2D. Furthermore, patients with severe acute/chronic diseases were excluded from the study, and this may lead to an underestimate of the proportion of patients who meet the dietary recommendations. Lastly, the detection of clinical and instrumental parameters of muscle mass, strength, and performance in relation to protein and leucine intake should add value to the observed data.

Future epidemiological studies and clinical trials are needed to investigate the impact of daily protein and leucine intake on objective variables of muscle health evaluated by imagine techniques, anthropometric parameters, body composition, muscle strength and physical performance, and plasma amino acid concentrations.

## 5. Conclusions

Our data indicate that in elderly patients with T2D, the average protein intake is low, particularly at breakfast and dinner, and that the leucine intake at each meal is remarkably below the recommendations. These data raise the need to implement nutritional strategies capable of increasing protein intake in the elderly with T2D in order to counteract aging-related skeletal muscle catabolism.

## Figures and Tables

**Figure 1 nutrients-15-01345-f001:**
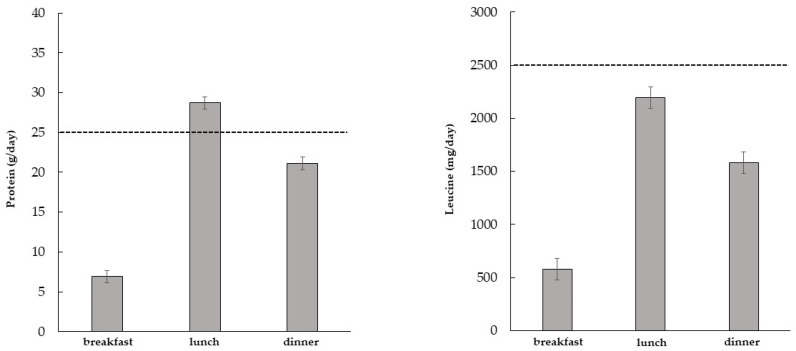
Mean intake of protein and leucine at main meals in elderly patients with type 2 diabetes. The dotted line represents the threshold of the recommended intake [16] in elderly people.

**Table 1 nutrients-15-01345-t001:** Main characteristics of the study population.

Variables	All Patients (n = 138)
Sex (men/women)	91/47
Age (years)	72 ± 4
BMI (kg/m^2^)	29.4 ± 4.7
Waist circumference (cm)	102 ± 11
Obese (%)	42%
Smoking (%)	24
Inactive * (%)	64
Duration of diabetes (years)	18 ± 9
HbA1c (%)	7.1 ± 1.0
HDL-Cholesterol (mg/dL)	49 ± 12
LDL-Cholesterol (mg/dL)	89 ± 29
Triglycerides (mg/dL)	124 ± 59
Hypertension (%)	86
Dyslipidemia (%)	91
Retinopathy (%)	21
Neuropathy (%)	23
Nephropathy (%)	24
Cardiovascular diseases (%)	36
Glucose lowering therapy	
- Metformin (%)	69
- DPP-4i (%)	28
- Sulfonylureas (%)	16
- Pioglitazone (%)	2
- GLP-1 RAs (%)	12
- SGLT2i (%)	21
- Insulin (%)	41
Anti-hypertensive therapy (%)	86
Lipid-lowering therapy (%)	85

Data are expressed as the mean ± standard deviation, or percentage. * According to the scoring system provided by the IPAQ scoring protocol. BMI: body mass index; HbA1c: glycated hemoglobin; DPP-4i: dipeptidyl peptidase-4 inhibitors; GLP-1 RAs: glucagon-like peptide-1 receptor agonists; SGLT2i: sodium-glucose transport protein 2 inhibitors.

**Table 2 nutrients-15-01345-t002:** Daily energy and nutrient intake of the study population.

Variables		All patients (n = 138)	Men (n = 91)	Women (n = 47)	*p*
Energy	(Kcal/day)	1414 ± 342	1462 ± 345	1322 ± 320	0.022
Protein	(g/kg BW/day)	0.9 ± 0.2	0.9 ± 0.2	1.0 ± 0.3	0.029
	(g/day)	57 ± 16	57 ± 16	55 ± 14	0.493
	(%TEI)	17 ± 2.9	17 ± 2.9	17 ± 2.9	0.232
Vegetal protein	(g/day)	25 ± 7.2	25 ± 7.3	23 ± 7.1	0.140
Animal protein	(g/day)	32 ± 14	32 ± 15	32 ± 12	0.962
Leucine	(mg/day)	4383 ± 1484	4348 ± 1292	4450 ± 1810	0.702
Isoleucine	(mg/day)	2447 ± 1113	2396 ± 740	2545 ± 1608	0.458
Valine	(mg/day)	2879 ± 1275	2826 ± 850	2981 ± 1842	0.500
Fat	(g/day)	53 ± 19	55 ± 20	50 ± 17	0.106
	(%TEI)	35 ± 6.5	35 ± 6.6	34 ± 6.2	0.509
Carbohydrates	(g/day)	167 ± 41	171 ± 40	159 ± 42	0.116
	(%TEI)	48 ± 7.2	48 ± 7.2	48 ± 7.3	0.808
Simple sugars	(g/day)	54 ± 20	53 ± 20	57 ± 21	0.237
Fibre	(g/day)	16 ± 6.1	16 ± 5.9	16 ± 6.5	0.570

Data are expressed as the mean ± standard deviation. g: grams; BW: body weight; TEI: total energy intake.

**Table 3 nutrients-15-01345-t003:** Daily energy intake and macronutrient composition of the main meals consumed by the study population.

Variables	Breakfast		Lunch		Dinner	
	Men (n = 91)	Women (n = 47)	Men (n = 91)	Women (n = 47)	Men (n = 91)	Women (n = 47)
Energy (Kcal/day)	175 ± 94	174 ± 81	763 ± 209	647 ± 210 **	510 ± 200	444 ± 161 *
Protein (g/day)	6.8 ± 3.7	7.1 ± 3.7	29 ± 11	28 ± 9.0	22 ± 11	20 ± 9.0
Vegetal protein (g/day)	2.7 ± 1.7	2.5 ± 1.4	14 ± 4.7	14 ± 4.4	8.0 ± 4.3	6.5 ± 3.6 *
Animal protein (g/day)	3.9 ± 2.9	4.5 ± 3.1	15 ± 10	14 ± 8.0	14 ± 9.9	13 ± 8.4
Fat (g/day)	4.2 ± 4.0	4.5 ± 3.8	29 ± 11	25 ± 12 *	22 ± 12	20 ±9.4
Carbohydrates (g/day)	29 ± 15	28 ± 11	87 ± 25	75 ± 24 **	51 ± 20	45 ± 22
Simple sugars (g/day)	13 ± 8.0	13 ± 7.0	20 ± 10	18 ± 11	16 ± 8.4	17 ± 9.7
Fibre (g/day)	1.1 ± 1.0	1.1 ± 1.1	8.2 ± 3.1	8.3 ± 4.0	5.9 ± 3.5	5.9 ± 2.8

Data are expressed as the mean ± standard deviation. * *p* < 0.05; ** *p* < 0.01. g: grams.

## Data Availability

Not applicable.

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
