# Peer review of "Protein and Leucine Intake at Main Meals in Elderly People with Type 2 Diabetes"

_nutrients, 2023, doi:10.3390/nu15061345_

Round 1
Reviewer 1 Report
1. Please the authors to further explain what is the scientific problem of the paper?
2. Introduction: please introduce the research purpose of the paper in detail. Especially, why chose leucine?
3. “None of the patients reached the recommended protein intake at breakfast”: Is there a problem with the author's selection for elderly people. For elderly people, some people should meet all the recommended standards. If not, is the conclusion reliable?
4. “Appropriate nutritional approaches focused on the quantity and quality of protein intake should be implemented in the elderly with T2D.”: What is the basis for this conclusion.
Author Response
We thank the reviewer for the positive comments. We have considered the various suggestions and have accordingly rewritten the manuscript. The changes in the new version of the text are written in red.
- Please the authors to further explain what is the scientific problem of the paper?
R: thank you for the suggestion! The aim of the paper is to provide additional information on the daily protein and leucine intake in elderly patients with T2D, considering that there is little evidence on the adherence of elderly people with T2D to the recommended protein intake, and, to the best of your knowledge, no data are available on the daily meal distribution of leucine intake -one of the most important branched-chain amino acids involved in muscle growth- in T2D patients who present a higher risk of loss of muscle mass.
Following reviewer’s comment, we have clarified this point in the revised paper as follows:
“Protein turnover at the level of skeletal muscle is regulated by several factors including the digestion of food proteins and the subsequent absorption of amino acids, the release of post-prandial insulin, the threshold of muscle anabolism, and the microvascular perfusion of the tissue [13]. Most of these processes can be greatly affected by the presence of diabetes, especially if in poor metabolic control. Elderly people with diabetes, in fact, present a greater risk of loss of muscle mass and strength and, consequently, a higher rate of physical disability [25]. Based on these considerations, they can greatly benefit from an adequate protein intake”, Lines: 64-71.
“In view of the important role of protein intake in relation to muscle mass, strength, and function in elderly people, and considering the importance of the content of branched-chain amino acids, leucine in particular for its powerful effect on muscle growth [16,21,22], the aim of the present study is to provide additional information on the daily protein and leucine intake in elderly patients with T2D.” Lines 77-80.
- Introduction: please introduce the research purpose of the paper in detail. Especially, why chose leucine?
R: As suggested, we have described the purpose of the research in more detail.
“Lastly, the quality of protein, the source (animal or plant) and the amino acids content, particularly the branched-chain amino acids, primarily leucine, are equally important for the skeletal muscle health [21,22]. As known, leucine beneficially acts on protein anabolism through the enzymatic activation of the Mammal Target of Rapamycin (mTOR), a serine-threonine protein kinase, involved in the regulation of cellular growth and protein turnover by promoting protein synthesis and inhibiting proteolysis [21,22].” Lines 56-62.
“In view of the important role of protein intake in relation to muscle mass, strength, and function in elderly people, and considering the importance of the content of branched-chain amino acids, leucine in particular for its powerful effect on muscle growth [16,21,22], the aim of the present study is to provide…” Lines 77-80
- “None of the patients reached the recommended protein intake at breakfast”: Is there a problem with the author's selection for elderly people. For elderly people, some people should meet all the recommended standards. If not, is the conclusion reliable?
R: In our population we have observed that in real life none of the elderly patients meet the recommended intake of protein at breakfast. This data is in accordance with evidence coming from epidemiological data showing that a significant percentage of individuals over 60 are more or less below the RDA as concerns specific food components and approximately 90% of elderly individuals do not assimilate the RDA for total calorie intake and 35% for protein intake. Furthermore, data presented in the Seneca study shows that in Italy the average percentage of calorie intake at breakfast is limited and lower compared to other European countries [Affinita, A., et al. Breakfast: a multidisciplinary approach. Ital J Pediatr 39, 44 (2013). https://doi.org/10.1186/1824-7288-39-44; Schlettweingsell D, et al. Meal patterns in the SENECA study of nutrition and the elderly in Europe: assessment method and preliminary results on the role of the midday meal. Appetite. 1999 Feb;32(1):15-22. doi: 10.1006/appe.1998.0191].
Furthermore, the breakfast is rarely high in protein [Cossu M, et al. A nutritional evaluation of various typical Italian breakfast products: a comparison of macronutrient composition and glycaemic index values. Int J Food Sci Nutr. 2018 Sep;69(6):676-681. doi: 10.1080/09637486.2017.1408060].
We have added this point in the revised manuscript, lines 241-243.
- “Appropriate nutritional approaches focused on the quantity and quality of protein intake should be implemented in the elderly with T2D.”: What is the basis for this conclusion.
R: Following reviewer ‘suggestion we have rephrased the conclusion of the abstract as follows: “These data raise the need to implement nutritional strategies capable of increasing protein and leucine intake in elderly with T2D”. Lines 27-28.

Reviewer 2 Report
As for the discussion, it is advisable to add that the loss of muscle mass may also be due to a decrease in their levels of physical exercise despite the fact that they consume the appropriate dose of protein.
Line 211-I would also like to point out that in order for protein intake to be effective, it is important to specify the optimal time of day according to their lifestyle, and that although the time of day for the highest protein intake may vary between cultures, its effectiveness is not affected because the work rhythms are also different.
As for the methodology, I find it to be scarce, although it is based on the objective of the study. In my experience, for future research, I would not rely solely on an observation sheet, but I would add some anthropometric control to get a good knowledge of their body condition. Similarly, I would add a strength test to check that the decrease in protein intake is associated with a real decrease in muscle mass and therefore muscle strength.
It should also be accompanied by a blood test to certify the internal body parameters.
Author Response
We thank the reviewer for the positive comments. We have considered the various suggestions and have accordingly rewritten the manuscript. The changes in the new version of the text are written in red.
- As for the discussion, it is advisable to add that the loss of muscle mass may also be due to a decrease in their levels of physical exercise despite the fact that they consume the appropriate dose of protein.
R: In agreement with the reviewer’s comment, we have added this point in the discussion.
“It is important to underline that muscle health is also influenced by physical activity, and in our population, the proportion of inactive patients was quite higher (64%). In this regard, evidence shows that protein supplementation increases muscle mass and handgrip strength in older adults only when combined with resistance exercise [Kirwan RP, Am J Clin Nutr. 2022,115(3):897-913], suggesting that in elderly people, according to functional status, other determinants such as regular resistance exercise and reduced sedentary time might positively impact muscle health beyond adequate protein intake [Peterson, M.D.; Med. Sci. Sports Exerc. 2011, 43, 249–258.64].” Lines 298-304.
- Line 211-I would also like to point out that in order for protein intake to be effective, it is important to specify the optimal time of day according to their lifestyle, and that although the time of day for the highest protein intake may vary between cultures, its effectiveness is not affected because the work rhythms are also different.
R: Thank you for the comment. We added the following point. “…considering that also lifestyle factors and physical activity are predictors of muscle health independently of daily protein distribution [Farsijani, S. et al. Am J Clin Nutr. 2016, 104, 694-703]” lines 247-248.
- As for the methodology, I find it to be scarce, although it is based on the objective of the study. In my experience, for future research, I would not rely solely on an observation sheet, but I would add some anthropometric control to get a good knowledge of their body condition. Similarly, I would add a strength test to check that the decrease in protein intake is associated with a real decrease in muscle mass and therefore muscle strength.
R: We thank the reviewer for his/her important suggestions. We have added a comment in the limitations of the study.
“Lastly, the detection of clinical and instrumental parameters of muscle mass, strength, and performance, in relation to protein and leucine intake should add value to the observed data. Future epidemiological studies and clinical trials are needed to investigate the impact of daily protein and leucine intake on objective variables of muscle health evaluated by imagine techniques, anthropometric parameters, and body composition, as well as the evaluation of muscle strength and physical performance together with the plasma concentration of amino acids”. Lines 317-324.
- It should also be accompanied by a blood test to certify the internal body parameters.
R: See response above.
